# Parameter identification of a fractional-order human-exoskeleton coupling model with the segmented form for the lower limb

1st Zhenlei Chen
*School of Automation Engineering*
*University of Electronic Science and Technology of China*
Chengdu, China
zhenlei_chen@std.uestc.edu.cn

2nd Tieshan Li
*School of Automation Engineering*
*University of Electronic Science and Technology of China*
Chengdu, China
tieshanli@126.com

3rd Yao Yan
*School of Aeronautics and Astronautics*
*University of Electronic Science and Technology of China*
Chengdu, China
y.yan@uestc.edu.cn

4th Qing Guo
*School of Aeronautics and Astronautics*
*University of Electronic Science and Technology of China*
Chengdu, China
guoqinguestc@uestc.edu.cn

*Abstract*—This paper presents a human-exoskeleton coupling model for estimating the human-exoskeleton coupling force. Specifically, to enhance the model's ability to characterize the complex mechanical characteristics of human-exoskeleton coupling, the fractional-order differential term is introduced. Moreover, the coupling model adopts a segmented form to characterize the different mechanical characteristics of front-side and back-side compression in the human-exoskeleton coupling. Using experimental data measured by the self-developed human-exoskeleton coupling measurement platform, the corresponding model parameters for 10 adult male and 10 adult female volunteers (Age: $23.65 \pm 4.03$ years, Height: $165.60 \pm 8.32$ cm, Weight: $62.35 \pm 14.09$ kg) were identified via Neighborhood Field Optimization (NFO) and Least Squares (LS) methods. Based on the analysis of the estimated root mean square error (RMSE) of the coupling force, it was verified that the coupling force can be estimated accurately based on the proposed fractional-order human-exoskeleton coupling model with the segmented form.

*Index Terms*—Human-exoskeleton coupling model, Fractional-order differential term, Segmented form, Model parameter identification

## I. INTRODUCTION

The exoskeleton is employed to facilitate human rehabilitation, strength augmentation, or locomotion assistance, combining human wisdom and robotic strength. Since 2000, exoskeleton technology has experienced rapid development, with several notable exoskeleton prototypes being designed, such as C-ALEX [1] and LOPES [2] for rehabilitation, HULC [3] and BLEEX [4] for strength augmentation, and ReWalk [5] and HAL [6] for locomotion assistance. The control mode of an exoskeleton can generally be classified into two groups: passive mode and active mode. An exoskeleton with passive mode drives the wearer to track the pre-generated desired trajectories [7], which are obtained from measurements of a healthy subject or designed by a professional sports planner. In active mode, the actual trajectory of the exoskeleton is adjusted in accordance with the wearer's motion intention [8]. Specifically for the position control of the exoskeleton, a variety of control algorithms are currently being studied, such as proportional-integral-derivative, fuzzy, adaptive, sliding-mode controllers [9]–[11].

To ensure the performance of human-exoskeleton coupling motion, it is necessary to investigate the motion characteristics of the human body. In recent years, some research has been conducted on the mechanical characteristic of human limbs in motion, including upper and lower limbs [12], [13]. Ma *et al.* [14] proposed an estimation model for the endpoint stiffness of the human arm from electromyographic signal (sEMG) and elbow angle. Liu *et al.* [15] combined muscle activation and muscle contraction dynamics to identify upper limb stiffness. For the limb-connected exoskeleton, to accurately perceive or estimate the wearer's motion intention, it is necessary to establish an accurate human-exoskeleton coupling model and identify its parameters. Yan *et al.* [16], [17] proposed a simple linear damping-spring human-exoskeleton coupling model, and the model parameters were identified through physical coupling experiments. Huang *et al.* [18] proposed a complex nonlinear human-exoskeleton coupling model for predicting human-exoskeleton coupling forces. This study aims to introduce the fractional-order differential term to enhance the capability to characterize the mechanical characteristic of human-exoskeleton coupling, and to utilize the segmented form for the asymmetry of human-exoskeleton coupling. The

This research is supported by National Natural Science Foundation of China (Grants No. 52305054, 51939001, 12072068, and 52175046), National Natural Science Foundation of Sichuan (Grant No. 2024NSFSC1479), China Postdoctoral Science Foundation (Grant No. 2022M720024), and Sichuan Science and Technology Program (Grant No. 24ZDYF0070). (Corresponding authors:Tieshan Li.)

fractional order calculus has been successfully utilized in modeling viscoelastic materials [19]. Specifically, Aydin *et al*. [20] proposed a fractional order admittance controller for physical human-robot interaction motion.

The current human-exoskeleton coupling model has issues with low accuracy or a complex form. Hence, a fractional-order human-exoskeleton coupling model with the segmented form is proposed. The main contributions of this study are as follows:

(1) In the proposed human-exoskeleton coupling model, the fractional-order differential term is introduced to enhance the capability to characterize the mechanical characteristics of human-exoskeleton coupling. Additionally, a segmented form is employed to characterize the mechanical characteristics in different human-exoskeleton coupling states (front-side and back-side compression).

(2) A human-exoskeleton coupling model parameter identification method based on the NFO and LS methods is introduced. The coupling model parameters of 20 volunteers were identified under different coupling positions and levels of looseness. To verify the accuracy of the proposed human-exoskeleton coupling model and the effectiveness of the parameter identification method, the RMSE of the coupling force was analyzed.

Since human body motion primarily occurs in the sagittal plane, and this study focuses on the human-exoskeleton coupling normal force caused by the lateral compression of the soft human-exoskeleton coupling (deformation and compression of human soft tissue caused by the lateral pulling of belt), only the human-exoskeleton coupling model along the $X$-axis is considered. Therefore, the coupling force and relative displacement mentioned in the following text refer specifically to the components along the $X$-axis (see Fig. 1).

The remainder of this paper is structured as follows. Section II introduces the self-developed human-exoskeleton coupling measurement platform and fractional order calculus. Section III describes the fractional-order human-exoskeleton coupling model with the segmented form and outlines the model parameter identification method. Section IV presents the results of the model parameter identification for 20 volunteers. Finally, conclusions are drawn in Section V.

## II. PLANT AND PRELIMINARY

### A. Human-exoskeleton Coupling Measurement Platform

For the purpose of identifying the parameters of the human-exoskeleton coupling model, it is imperative to accurately collect and record the motion displacement of human and coupling force during the human-exoskeleton coupling motion. Hence, we developed the human-exoskeleton coupling measurement platform shown in Figs. 2-3. The main structure of the device is a stainless steel rigid frame, which is considered an analog for the lower limb exoskeleton. A three-dimensional (3D) force sensor (LH-SZ-02-100) and two laser displacement sensors (ILD1420-200) are used, and the measured data are recorded at a sampling frequency of 15625 Hz and transferred to NI-DAQ Express using the NI USB-6210 device. Then, the

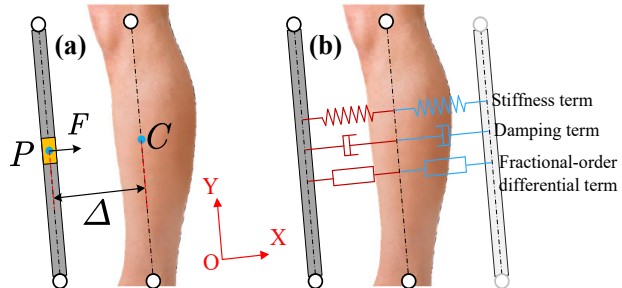

Fig. 1. Schematic diagram of human-exoskeleton coupling motion:(a) relative displacement $\Delta$ and coupling force $F$ at the coupling point, (b) the fractional-order human-exoskeleton coupling model with the segmented form

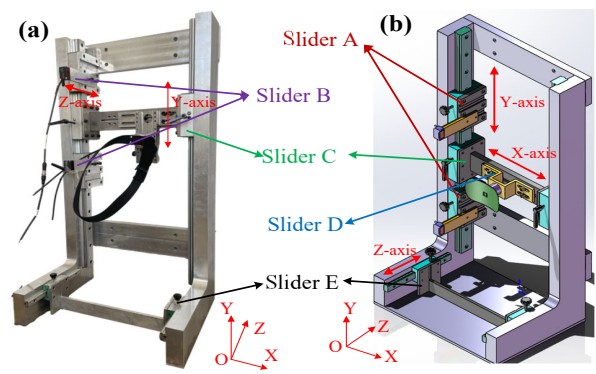

Fig. 2. Mechanical structure of the human-exoskeleton coupling measurement platform: (a) the photo, (b) the 3D design

actual measured data is filtered using a Butterworth lowpass filter (cutoff frequency:10 Hz) to remove measurement noise. Additionally, the position of these sensors can be adjusted by moving the designed sliders, allowing for adaptation to different volunteers. For instance, Sliders C and D facilitate the adjustment of the force sensor along the $Y$- and $X$-directions, while the laser sensors can be moved along the $Y$- and $Z$-directions by adjusting Sliders A and B, respectively.

### B. Fractional Order Calculus

Fractional order calculus is a generalization of integer order calculus, which allows integration and differentiation of non-integer orders. Several definitions exist for the fractional-order differintegral operator, including the Grunwald-Letnikov, Riemann-Liouville, and Caputo definitions.

In this study, we use the Simulink module from the FOMCON toolbox in Matlab/Simulink to perform fractional order derivative calculations. Then, a fractional-order human-exoskeleton coupling model with the segmented form is proposed to characterize the coupling characteristics between the human and exoskeleton.

## III. METHODS

### A. Human-exoskeleton Coupling Model

In the human-exoskeleton coupling collaborative task, the operator's lower limbs and the exoskeleton are bundled and connected using belts or other devices. The rigid exoskeleton,

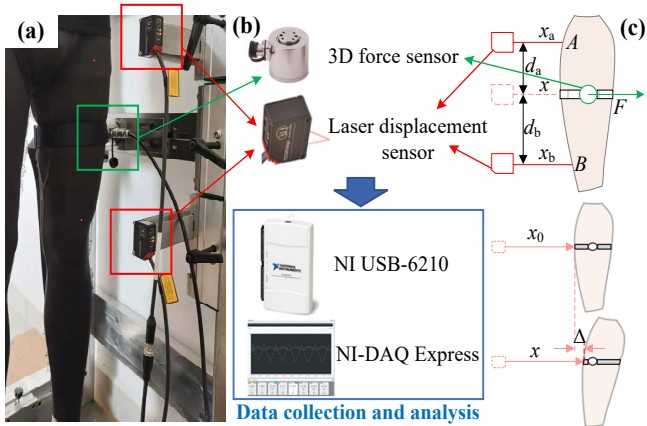

**Human-exoskeleton coupling measurement platform**

(a) (b) (c)

3D force sensor

Laser displacement sensor

NI USB-6210

NI-DAQ Express

**Data collection and analysis**

Fig. 3. Schematic diagram of the human-exoskeleton coupling measurement platform:(a) actual experimental scene, (b) sensors, data collection and analysis system, (c) calculation and measurement methods for the human-exoskeleton coupling relative displacement $\Delta$ and the coupling force $F$

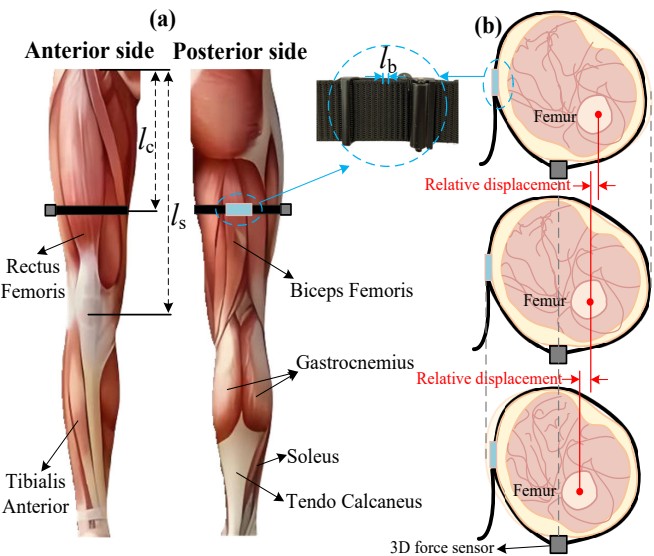

Fig. 4. (a) Anterior and posterior muscle distribution of the human lower limb, (b) Top view of the coupling position between the human and the exoskeleton in three states: front-side compression, human-exoskeleton synchronization, and back-side compression

constructed from metal or composite materials, is a typical rigid body. Similarly, the human skeleton, with an elasticity modulus ranging from 12 to 20 GPa [21], can also be regarded as a rigid body. The human-exoskeleton relative motion leads to compression of the soft human-exoskeleton coupling, including belt, skin, muscle, and fat, resulting in the human-exoskeleton coupling force. Therefore, to accurately infer or estimate the operator's motion intention based on the real-time coupling force, it is necessary to construct a sufficiently accurate human-exoskeleton coupling model.

As depicted in Fig. 1-(a), the relative motion between the coupling points $P$ on the exoskeleton and $C$ on the human leg results in the generation of the coupling force $F$. A simple stiffness-damping model for $F$ is expressed as

$$F = b_{sd}\dot{\Delta} + k_{sd}\Delta, \qquad (1)$$

where $\dot{\Delta}$ and $\Delta$ are the relative velocity and displacement at the human-exoskeleton coupling point, and $b_{sd}$ and $k_{sd}$ are the coupling damping and stiffness.

However, based on the following two considerations, it is challenging to accurately characterize the actual human-exoskeleton coupling characteristics using (1).

*1) Complex mechanical characteristics of human-exoskeleton coupling:* The human-exoskeleton coupling is composed of textile fabrics, including belt and pants, as well as human soft tissues, such as the fat layer, muscle, and skin of the human lower limb. As a result, it exhibits complex mechanical characteristics, such as viscoelasticity.

*2) Asymmetry of human-exoskeleton coupling:* The relative motion between the human lower limb and the exoskeleton results in the generation of human-exoskeleton coupling force, which is attributed to human-exoskeleton coupling compression. There is a distinct difference in the soft tissue on the front and back sides of the human leg, such as muscle distribution (see Fig. 4-(a)) and fat thickness. Therefore, there is a significant difference in the mechanical characteristics of

human-exoskeleton coupling between the two cases of front-side compression and back-side compression. In summary, there is a notable asymmetry in the mechanical characteristics of human-exoskeleton coupling.

Based on the aforementioned analysis, a fractional-order human-exoskeleton coupling model with the segmented form (see Fig. 1-(b)) is constructed, which is expressed as follows:

$$F = \begin{cases} b_{sfd-p}\dot{\Delta} + c_{sfd-p}\mathscr{D}^{\alpha}\Delta + k_{sfd-p}\Delta, & \Delta > 0 \\ b_{sfd-n}\dot{\Delta} + c_{sfd-n}\mathscr{D}^{\alpha}\Delta + k_{sfd-n}\Delta, & \Delta \leq 0 \end{cases}, \quad (2)$$

where $b_{sfd-i}$ and $k_{sfd-i}$ $(i = n, p)$ are the coupling damping and stiffness, $\mathscr{D}^{\alpha}\Delta$ is the fractional derivative of $\Delta$ with order $\alpha \in (0, 1)$, and $c_{sfd-i}$ $(i = n, p)$ is the coefficient of $\mathscr{D}^{\alpha}\Delta$. Generally, $\alpha$ is a pre-set parameter, and it is set to 0.5 for the actual model parameter identification in this study.

Compared to the stiffness-damping model (1), the proposed coupling model (2) employs a segmented form, using two sets of model parameters $(b_{sfd-i}, k_{sfd-i}, c_{sfd-i}, i = n, p)$ to characterize the mechanical characteristics in different human-exoskeleton coupling states (front-side compression $\Delta \leq 0$ and back-side compression $\Delta > 0$), where human-exoskeleton synchronization is treated as a special state of front-side compression. Additionally, the fractional-order differential term $c_{sfd-i}\mathscr{D}^{\alpha}\Delta$ $(i = n, p)$ is introduced to enhance the capability to characterize the mechanical characteristics of human-exoskeleton coupling.

### B. Parameter Identification of Coupling Model

During the human-exoskeleton coupling motion experiment, the volunteers were secured to the human-exoskeleton coupling measurement platform, yielding relative movement and coupling force. Furthermore, due to the elasticity of human-exoskeleton coupling, there exists local tissue deformation

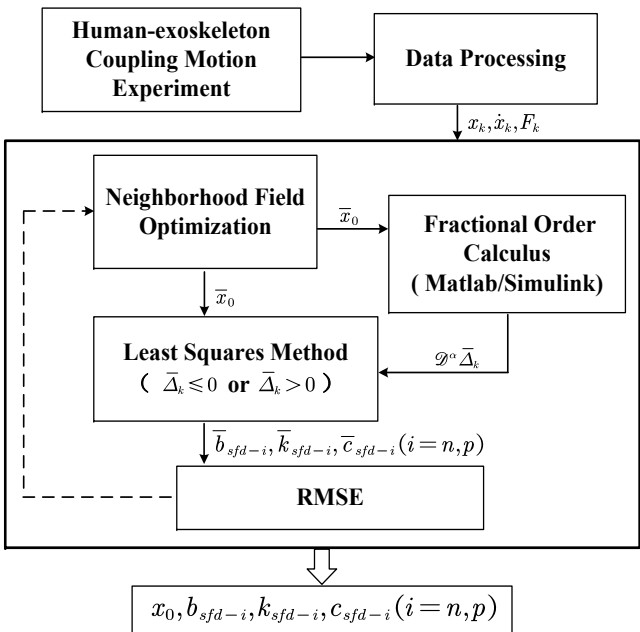

Fig. 5. Parameter identification flow chart of the fractional-order human-exoskeleton coupling model with the segmented form (2)

at the coupling point, which implies that the horizontal displacement of the coupling point cannot be directly measured. As shown in Fig. 3-(c), two independent laser sensors are employed to measure the horizontal displacements $x_a(t)$ and $x_b(t)$ of the measurement points $A$ and $B$, respectively. An indirect method is adopted to calculate the relative displacement $\Delta(t)$, which can be expressed as

$$\Delta(t) = x(t) - x_0, \quad x(t) = \frac{x_b(t)d_a + x_a(t)d_b}{d_a + d_b}, \quad (3)$$

where $x(t)$ is the horizontal displacement between the coupling point and the connection line of the dual laser sensors, and $x_0$ is the constant displacement for zero coupling force in the human-exoskeleton stationary state.

Actually, volunteers vary in leg shape, and their standing positions also differ across group experiments. Even within a single group experiment, the standing position of volunteers may slightly change. Hence, accurately measuring $x_0$ is difficult. In this study, $x_0$ is also considered a parameter to be identified, similar to $b_{sfd-i}$, $k_{sfd-i}$, and $c_{sfd-i}$ ($i = n, p$).

During the experiment, a series of data regarding the variables $x$ and $F$ were obtained, as follows:

$$\begin{cases} x_k = x(k \times \Delta t), k = 1, 2, 3, \ldots, k_{\max} \\ F_k = F(k \times \Delta t), k = 1, 2, 3, \ldots, k_{\max} \end{cases}, \quad (4)$$

where $\Delta t$ is the actual sampling period, and $k_{\max}$ is the number of data points. To balance accuracy and computational efficiency, the original data sequence was uniformly sampled at a ratio of 1 : 10 before parameter identification. Thus, the actual values of $\Delta t$ and $k_{\max}$ are $\frac{2}{3125}$ s and 93125. The relative velocity $\dot{x}_k$ used in parameter identification was approximately obtained by the backward difference method.

As shown in Fig. 5, during the process of parameter identification, the NFO algorithm is employed to optimize $x_0$, where NFO is a heuristic algorithm inspired by the attraction and repulsion between natural particles and their neighbors. In the NFO algorithm, when an individual adopts the value $\bar{x}_0$ in the optimization process (where $\bar{\bullet}$ represents the temporary value of $\bullet$ for an individual in a particular process), the corresponding fractional differential term $\mathscr{D}^\alpha \bar{\Delta}_k = \mathscr{D}^\alpha \bar{\Delta}(k \times \Delta t)$ ($k = 1, 2, 3, \ldots, k_{\max}$) is calculated using Matlab/Simulink. Then, to ensure the used data points of fractional differential term series reach steady state, the first $k_{\min} = 19999$ data points in all data series are ignored, and the data points $\mathscr{D}^\alpha \bar{\Delta}_k$, $\bar{\Delta}_k$, $\dot{\Delta}_k$, and $F_k$ ($k = k_{\min} + 1, k_{\min} + 2, \ldots, k_{\max}$) are used for parameter identification, where $\bar{\Delta}_k = x_k - \bar{x}_0$ and $\dot{\Delta}_k = \dot{x}_k$. Based on the value of $\bar{\Delta}_k$ ($\bar{\Delta}_k \leq 0$ or $\bar{\Delta}_k > 0$), the data are divided into two groups to identify model parameters $\bar{b}_{sfd-i}$, $\bar{k}_{sfd-i}$, and $\bar{c}_{sfd-i}$ ($i = n, p$) using the LS method. The corresponding estimated coupling force $\bar{F}_k^{\text{fit}}$ is calculated according to (2). The optimization object of NFO is to minimize the estimated RMSE of the coupling force. The RMSE is defined as

$$\text{RMSE} = \sqrt{\frac{1}{k_{\max} - k_{\min}} \sum_{k=k_{\min}+1}^{k_{\max}} (\bar{F}_k^{\text{fit}} - F_k)^2}. \quad (5)$$

When the NFO algorithm reaches the termination condition, the optimization value of $x_0$ and the corresponding parameters $b_{sfd-i}$, $k_{sfd-i}$, and $c_{sfd-i}$ ($i = n, p$) are the parameter identification results.

### C. Coupling Position and Looseness

For this study, a total of 20 adult volunteers, consisting of 10 males and 10 females with varying body types (Age: $23.65 \pm 4.03$ years, Height: $165.60 \pm 8.32$ cm, Weight: $62.35 \pm 14.09$ kg), were recruited. The volunteers operated the platform with the human-exoskeleton coupling position and tightness as control variables.

As shown in Fig. 4-(a), considering differences in leg length among volunteers, a length ratio $l_p$ is used to represent the coupling position and is defined as

$$l_p = \frac{l_c}{l_s} \times 100\%, \quad (6)$$

where $l_c$ is the vertical distance from the proximal end of each segment to the coupling position, and $l_s$ is the segment length.

Similarly, leg circumference at the same coupling position varies significantly among volunteers. As shown in Fig. 4, the canvas belt has evenly distributed teeth with an interval of $l_b = 3.53$ mm, and the coupling tightness can be adjusted by maneuvering and securing the box-frame buckle into place.

For the initial test of each volunteer, the belt was tightened to the maximum extent that the volunteer could tolerate. Subsequently, for every subsequent test, the coupling was loosened by 2 teeth. Therefore, the releasing length of the belt is used to quantify the degree of looseness, and the looseness $L_{\text{looseness}}$ is defined as

$$L_{\text{looseness}} = l_b n_s, \quad (7)$$

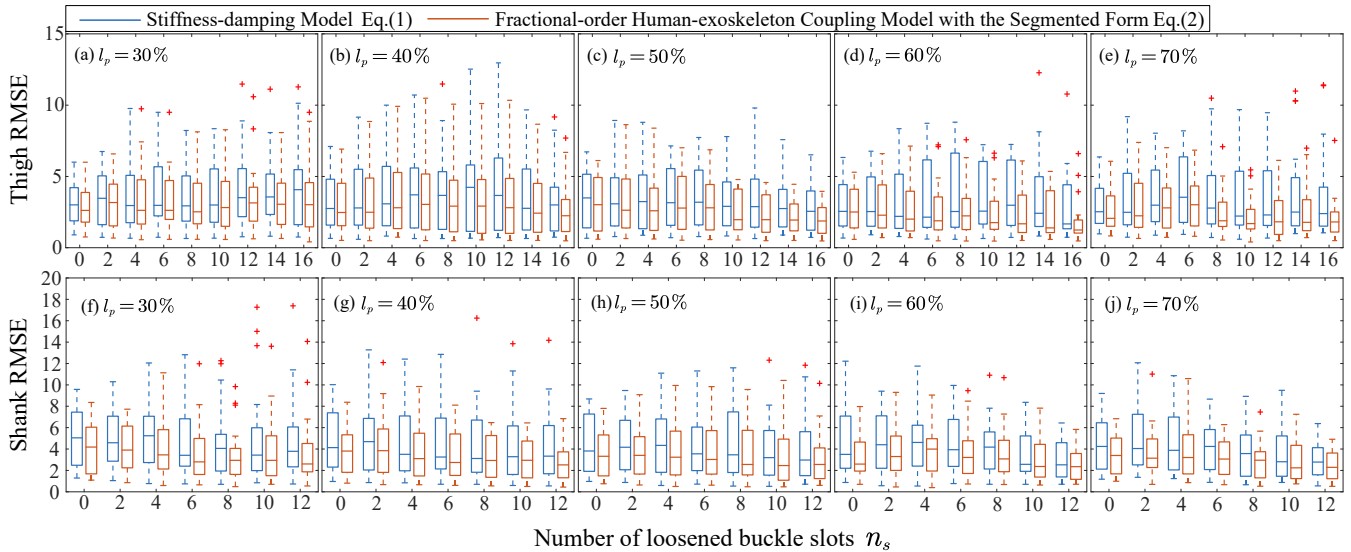

Fig. 6. The RMSE values of the human-exoskeleton coupling force based on stiffness-damping model (1) and the fractional-order human-exoskeleton coupling model with the segmented form (2) are shown. The level of looseness indicated on the horizontal axis is represented by the number of belt slots loosened by the buckle, $n_s$.

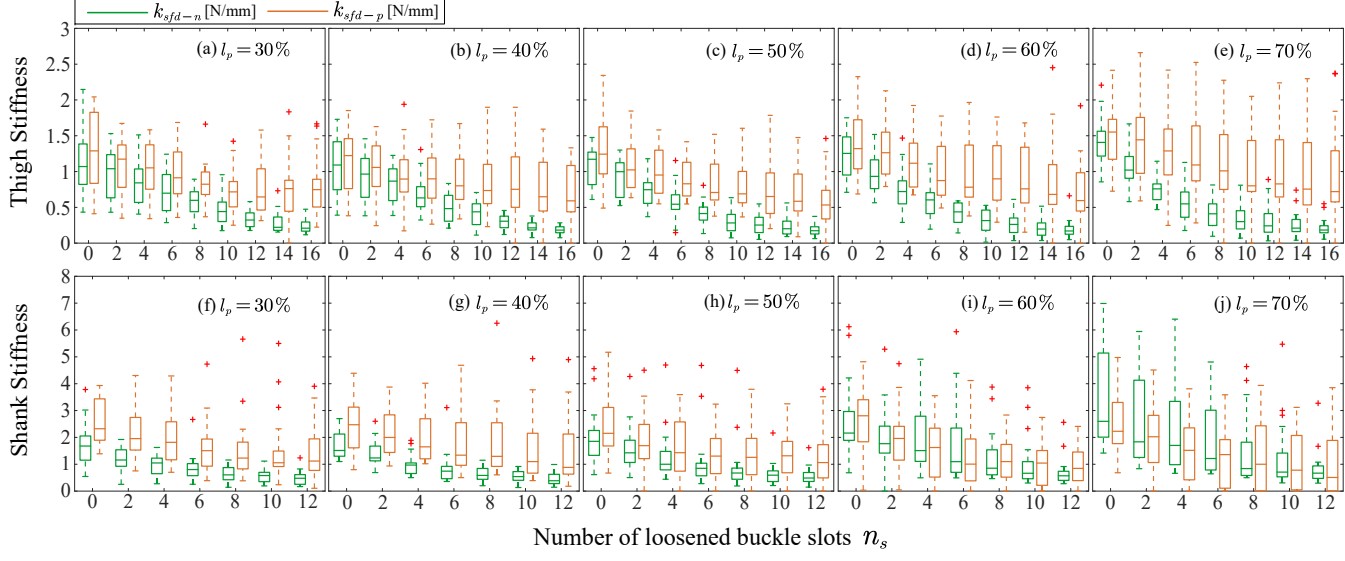

Fig. 7. The human-exoskeleton coupling stiffness parameters $k_{sfd-n}$ and $k_{sfd-p}$ are shown. The level of looseness indicated on the horizontal axis is represented by the number of belt slots loosened by the buckle, $n_s$.

where $n_s$ is the number of buckle slots to be loosened.

## IV. RESULTS AND DISCUSSION

For each volunteer, the belt was gradually loosened to 9 and 7 levels of tightness in the thigh and shank, respectively. At each loosened level, the coupling parameters were identified for $l_p = 30\%$, $40\%$, $50\%$, $60\%$, and $70\%$. This resulted in 45 and 35 groups of data for identifying thigh- and shank-exoskeleton coupling model parameters for each volunteer, with the fractional order $\alpha = 0.5$.

In this section, the boxplot is adopted to illustrate the statistical results, where outliers are identified as data points

that lie beyond 1.5 times the interquartile range from the median. The interquartile range is defined as the range between the 25th and 75th percentiles of the data. For the stiffness-damping model (1), the corresponding model parameters ($b_{sd}$, $k_{sd}$, $x_0$) were also identified using NFO and LS methods, and the identification process is similar to the one proposed in this study. As shown in Fig. 6, based on the stiffness-damping model (1) and the fractional-order coupling model with the segmented form (2), each boxplot indicates the distribution of RMSE, calculated from the final parameter identification results using (5), of the force-fitted among 20 volunteers at a particular coupling position $l_p$ and with the same level of

coupling looseness. For comparison, the estimation of human-exoskeleton coupling force by the proposed fractional-order coupling model with the segmented form (2) exhibits better accuracy and a more concentrated distribution.

Given that stiffness is the most prominent mechanical characteristic of human-exoskeleton coupling, we specifically analyzed the identified stiffness coefficients $k_{sfd-n}$ and $k_{sfd-p}$. Fig. 7 shows the influence of coupling looseness on stiffness in two human-exoskeleton coupling states ($\Delta \leq 0$ or $\Delta > 0$), represented by $k_{sfd-n}$ and $k_{sfd-p}$, across 10 different coupling positions. Regardless of the coupling position $l_p$, the values of $k_{sfd-n}$ and $k_{sfd-p}$ generally exhibit a downward trend as the looseness increases. Because it is more difficult to maintain balance when the leg leans backward compared to forward during the experiment, there may be a lack of data for backward leg leaning in certain experiments for some volunteers. In rare cases, there may also be a lack of data for forward leg leaning. Furthermore, under the same $l_p$ and looseness, the value of $k_{sfd-p}$ is generally larger than that of $k_{sfd-n}$ in most cases. When $\Delta > 0$, this indicates that there is a squeeze between the back side of the human leg and the belt. Conversely, when $\Delta \leq 0$, it signifies that the front side of the human leg is squeezed by the belt. As shown in Fig. 4-(a), whether in the thigh or shank, the posterior muscle tissue is typically thicker and more developed than the anterior. In addition, the muscle tissue of the lower limbs remained active throughout most of the experiment. Therefore, in general, it is reasonable to conclude that $k_{sfd-p}$ is larger than $k_{sfd-n}$.

## V. Conclusion

This paper proposes a fractional-order human-exoskeleton coupling model with the segmented form (2) to represent the relationship between the human-exoskeleton relative motion and coupling force. Firstly, for the human-exoskeleton coupling, which includes components such as belt and human soft tissue, the fractional-order differential term is introduced to enhance the capability to characterize the mechanical characteristics. In addition, the segmented form is adopted for the asymmetry of human-exoskeleton coupling. Based on the experimental data from 20 volunteers in the human-exoskeleton coupling motion experiment, the coupling model parameters in (2) were identified using NFO and LS methods. Then, by analyzing and discussing the results of model parameter identification, it was verified that the accuracy of the proposed fractional-order human-exoskeleton coupling model with the segmented form (2) is generally superior to that of the stiffness-damping model (1).

In future research, we will use the proposed fractional-order human-exoskeleton coupling model with the segmented form (2) to estimate real-time human motion intention through coupling force and to control a lower limb exoskeleton.

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
