# OpenReview forum: "Parameter identification of a fractional-order human-exoskeleton coupling model with the segmented form for the lower limb"
_IEEE.org/ICIST/2024/Conference — IEEE ICIST 2024 Conference Submission_

### Official Review · Reviewer_Lvwr · 2024-08-21
**This paper established a novel human-exoskeleton coupling dynamic model for estimating human-exoskeleton coupling forces. Meanwhile, it can be verified that the coupling force can be estimated accurately based on the proposed fractional-order human-exoskeleton coupling dynamic model with segmented form. This paper presents an interesting approach. Here are some comments.**

**Rating:** 10
**Confidence:** 5

**Review:**

Question 1:
Please elaborate on the specific advantages of introducing fractional-order differential terms in enhancing the characterization of human-exoskeleton coupling dynamics? How does this approach compare with traditional integer-order differential terms?
Question 2:
Regarding the proposed human-exoskeleton coupling model parameter identification method based on NFO and LS, could the authors provide clarity on how these methods were integrated and applied to identify coupling model parameters across different coupling positions and levels of looseness? Additionally, were there any challenges encountered during the parameter identification process?
Question 3:
In the analysis of the RMSE of the coupling force to validate the proposed model and parameter identification method, could the authors discuss the statistical significance of the results obtained from the 20 volunteers? How robust are the findings across varying experimental conditions and participant demographics?

---

### Official Review · Reviewer_FCwH · 2024-08-21
**This paper has a clear logic and is innovative. The experimental results are convincing. It is strongly recommended to be published in IEEE ICIST 2024.**

**Rating:** 10
**Confidence:** 5

**Review:**

This paper has a clear logic and is innovative. The experimental results are convincing. Here are some reference comments.

How does the introduction of the fractional-order differential term specifically enhance the characterization of dynamic characteristics in the human-exoskeleton coupling model? Please provide some theoretical insights or mathematical justifications to support your claim.
In the proposed segmented form for different coupling states (front-side compression vs. back-side compression), how are the transitions between these states handled or identified in the model? Are there any assumptions or criteria employed to determine when a state transition occurs?

What are the limitations of the proposed NFO and LS-based parameter identification method? Are there any potential biases or inaccuracies that could arise from the method, and how were these addressed or minimized in your experiments? Also, how sensitive are the identified coupling model parameters to changes in the experimental conditions (e.g., variations in subjects, coupling positions, or looseness levels)?

---

### Official Review · Reviewer_2hvJ · 2024-08-24
**It can be accepted.**

**Rating:** 8
**Confidence:** 4

**Review:**

This paper established a novel human-exoskeleton coupling dynamic model for estimating human-exoskeleton coupling forces. The results obtained are valuable and could be accepted if the following issues are clarified: (1) The introduction should include a more detailed summary of the limitations in the relevant studies. (2) The study's structure should be briefly summarized at the end of Section 1. (3) More detailed analysis in the simulation section is advised to better interpret the main results. (4) There are a number of spelling and grammatical errors that need to be carefully reviewed and polished.

---

### Decision · Program_Chairs · 2024-09-08

Accept (Oral)